# Supported employment interventions for workplace mental health of persons with mental disabilities in low-to-middle income countries: A scoping review

Edwin Mavindidze[1]*, Clement Nhunzvi[1], Lana Van Niekerk[2]

1 Faculty of Medicine and Health Sciences, Department of Primary Health Care Sciences, Occupational Therapy Programme, Rehabilitation Sciences Unit, University of Zimbabwe, Harare, Zimbabwe, 2 Faculty of Medicine and Health Sciences, Division of Occupational Therapy, Stellenbosch University, Cape Town, Western Cape, South Africa

◉ These authors contributed equally to this work.
* eddie.mavie@gmail.com

## Abstract

### Objective

To review the evidence of supported employment interventions in low-to-middle income countries, documents their impact for persons with mental disorders in the open labour market and well as support decision making for its wider implementation in the workplace.

### Design

The scoping review is conducted following guidelines in the Arksey and O'Malley (2005) Framework.

### Data sources and eligibility

Eleven databases which are PubMed, Scopus, Academic Search Premier, the Cumulative Index to Nursing and Allied Health Literature, Africa-Wide Information, Humanities International Complete, Web of Science, PsychInfo, SocINDEX, Open Grey and Sabinet were searched for articles published between January 2006 and January 2022. Both peer-reviewed articles and grey literature were eligible if they were on supported employment interventions in low-to-middle income countries. Only articles published in English were included.

### Study appraisal and synthesis

Articles were screened at title, abstract and full article levels by two independent teams with the use of Rayyan software. Deductive thematic analysis was used to synthesize evidence on the supported employment interventions implemented in LMICs, capturing evidence of their outcomes for persons with mental disabilities securing competitive work.

**Data Availability Statement:** All relevant data are within the paper and its Supporting Information files.

**Funding:** This research was commissioned by the Wellcome Trust as part of the Workplace Mental Health Commission 2021 (https://wellcome.org/reports/where-next-workplace-mental-health) awarded to EM(Team lead), CN and LvN. For the purpose of open access, Wellcome has applied a CC BY public copyright licence to any Author Accepted Manuscript version arising from this submission. The funders had no role in study design, data collection and analysis, decision to publish, or preparation of the manuscript.

**Competing interests:** The authors have declared that no competing interests exist.

## Results

The search yielded 7347 records and after screening by title and abstract, 188 studies were eligible for full article screening. Eight studies were included in this scoping review. Thematic descriptions of the findings were based on the availability of supported employment interventions within the context, the type of supported employment interventions as well as mental health and vocational outcomes in the workplace.

## Conclusions

There is limited evidence of supported employment interventions in low-to-middle income countries despite the promising potential it has as an intervention to address mental health problems in the workplace and facilitate work participation by persons with mental disabilities.

## Introduction

Work and employment are central tenets and expectations of adult life. However, for individuals with lived experience of mental disorders decent work continues to be an evading target. This is despite policy and legislative efforts at both international and local levels to promote equity in terms of rights to work for persons with disabilities including persons with mental disabilities [1,2]. In low-to-middle income countries (LMICs) persons with disabilities, including persons with mental disabilities, experience exclusion from participation in work [3–5]. Multiple barriers to attaining, retaining, maintaining and returning to work prevent persons with mental disabilities from participating in and experiencing decent work [4,6–8]. Persons with mental disabilities experience difficulties participating in the workplace due to dysfunctions associated with mental disorders [9]. Limited education, limited work experiences, recurring and persisting symptoms resulting in long periods of absence from work are significant personal barriers to employment [6,9]. Furthermore, community and systemic barriers like social exclusion, stigma, policy disincentives, inaccessible work rehabilitation services and unaccommodating work environments deter job seeking and thriving in the work place [9–11].

Work has a positive therapeutic effect for persons with mental disabilities and, despite the stress related to work demands, employment promotes good mental health and well-being [12]. Employment status has been identified as a social determinant of mental health and has an influence on mental health outcomes [13–16]. Mental health outcomes do not only imply changes in associated symptoms of mental disorders, but also include broader social functioning, community belonging, independence, relationships, recovery and quality of life [17]. Moreover, participation in work provides income required to access other social determinants of health, including access to health services and healthy daily living conditions. Strong and consistent evidence confirms the relationship between unemployment and ill-health, including an increased risk of mental disorders and adverse mental health outcomes like depression and anxiety [14,16] while employment has been related to reduced anxiety and depression [13]. More recently, a number of studies explored health outcomes associated with participation in work, contributing to growing evidence supporting positive associations between mental health and participation in work. A systematic review [18] undertaken to summarise best evidence on the "health effects of employment" (p. 730) reported strong evidence for employment

improving general mental health and reducing the risk for depression. Another systematic review comprising four studies of moderate quality focussed on mental health in considering the benefits of work [19]; it confirmed the value of work for employees' mental health. Return to work or re-employment have also been shown to improve mental health. Fifteen of eighteen studies included in a systematic review reported that health benefits were associated with work; either because health improved significantly on return to work or decreased significantly when work was lost [20].

Supported employment interventions are a group of work rehabilitation interventions that aim to improve access and participation in the open labour market for persons with mental disabilities [21]. Supported employment interventions uses a place-and-train model versus the more traditional train-and-place model. A place-and-train model entails putting service users in real life work situations without prior preparation and allowing them adapt to the workplace while providing them with ongoing support while they are in the workplace [22,23], while in the train-and-place model the individual is first taken through pre-vocational training to learn basic work habits and specific work skills before seeking employment [22]. A systematic review that included eight sources in which 'place-and-train' interventions were used concluded that re-employment utilising this model had a modest effect on quality of life [24].

Supported employment interventions are offered using various models. The most common model among supported employment interventions is the Individual Placement and Support (IPS) model [23,25], IPS focuses on ensuring rapid job search and placement of clients in competitive work settings of their choice, while continuing to provide support to accessing clinical services and counselling for an unlimited time [23]. Other models of supported employment interventions, such as Integrated Supported Employment (ISE), follow similar principles as IPS, and either combine IPS with other psychological interventions or embed it into another models [25–27]. ISE integrates occupational therapy services together with supported employment interventions which can take various forms including embedding work-related social skills training (WSST) into an IPS model [25,26]. WSST, IPS and ISE are defined in Table 1.

In other cases, supported employment interventions are embedded in another psychosocial rehabilitation model such as the Clubhouse model. Clubhouse supported employment provides competitive job placements with supports from the clubhouse both at work and away from work [30]. The Clubhouse model is a psychosocial rehabilitation model, were persons

**Table 1. Supported employment interventions in LMIC.**

| Supported Employment Model | Definition |
|---|---|
| Work-related Social Skills Training (WSST) | "a structured program to teach participants job interview skills, basic conversation and social survival skills for effective communication with supervisors, co-workers and customers. . . training on verbal and non-verbal communication, accurate social perception, assertiveness, grooming and personal appearance, greetings and other basic conversation skills. . . core work-related skills including those required for job searches, phone and face-to-face interviews, social skills in specific situations in the workplace like handling conflicts and requesting sick leave, and also problem-solving skills."[28] |
| Individual Placement and Support (IPS) | "[an] approach to vocational rehabilitation [which] incorporates eight principles: eligibility based on consumer choice, focus on competitive employment (i.e., jobs in integrated work settings in the competitive job market at prevailing wages with supervision provided by personnel employed by the business), integration of mental health and employment services, attention to client preferences, work incentives planning, rapid job search, systematic job development, and individualized job supports."[29] |
| Integrated Supported Employment (ISE) | combines IPS with WSST and other forms of work-related occupational therapy interventions [27,28] |

with mental disabilities intentionally come together, forming a therapeutic working community, to support each other beyond clinical outcomes [30].

Supported employment interventions have been implemented, extensively researched and well documented around the world [23,31–38]. Findings from a randomised control trial of two supported employment intervention programmes concluded that remunerated part-time work, undertaken within the Clubhouse-model, was found to improve quality of life and self-esteem for adults with mental illness [39]. Given reports of its effectiveness in high income settings like USA, UK, Australia, Sweden and Japan [32,40–43], supported employment interventions have the potential to improve work participation and engagement outcomes for persons with mental disabilities in LMICs. However, implementation of supported employment interventions in LMICs has been hindered by socioeconomic and sociopolitical factors in the various countries [44–47]. Progressive disability policies, founded on a human rights approach, are generally in place, but with little evidence of implementation [2]. Using Zimbabwe as an example it is clear that socioeconomic hardships that have persisted since the early 2000s resulted in high unemployment rates. Furthermore, sociocultural factors such as stigma and misconceptions of mental illness continue to hinder implementation of supported employment interventions in Zimbabwe. However, in recent years there have been policy changes in relation to the welfare of persons with disabilities, including severe mental disorders [1]. The Constitution of Zimbabwe most notably promotes the establishment of programmes that facilitate participation of persons with disabilities, both physical and mental, in work and is a key facilitator for implementation of supported employment interventions in Zimbabwe [1,6]. It should also be noted that implementation supports, such as providing funding, personnel training and legislative provisions, are a key ingredient to successful implementation of supported employment interventions [48]. Moreover, given the technical nature of supported employment interventions providing necessary support ensures access of the most appropriate and relevant service to persons with mental disabilities [48].

Despite all the research done there remains limited knowledge of supported employment interventions in LMICs. Although a scoping review has synthesised the existing evidence of vocational rehabilitation interventions being offered at institutions in LMICs [49], there is a dearth of evidence on support employment interventions in LMICs. This review, therefore, seeks to synthesize the existing literature pertaining to supported employment interventions in LMICs and document their impacts on persons with mental disabilities in the open labour market. Findings may be used to support decision makers in the development, implementation, and execution of supported employment interventions.

## Methodology

The scoping review, guided by Arksey and O'Malley framework [50] was done between November 2021 and February 2023. The purpose of the scoping review was to synthesize existing research from LMICs on supported employment interventions and gauge its potential in facilitating employment for persons with mental disabilities in LMICs. The five stages of the Arksey and O'Malley [50] and how they guided this scoping review are outlined below.

### Stage 1: Identifying the research questions

This scoping review aimed to answer two questions; "What is the existing literature on supported employment interventions for persons with mental disorders in LMIC?" and "What is the impact of supported employment interventions for persons with mental disabilities in LMIC?"

## Stage 2: Identifying relevant studies

We designed a search strategy and with assistance from a subject librarian selected databases and the selection criteria. The search was conducted, between July and October 2022, on 11 databases which are PubMed, Scopus, Academic Search Premier, the Cumulative Index to Nursing and Allied Health Literature (CINAHL), Africa-Wide Information, Humanities International Complete, Web of Science, PsychInfo, SocINDEX, Open Grey and Sabinet. Key search terms included supported employment and mental disorders and their alternate terms or derivatives (Table 2). We conducted a preliminary search on PubMed which allowed us to further refine our search terms, including MeSH terms. The search strategy was adapted accordingly for all the databases.

## Stage 3: Study selection

In this stage reviewers aimed to identify articles on supported employment interventions for mental disorders in LMICs, as well as articles on the efficacy of these supported employment interventions in LMICs. LMICs were defined according to the World Bank Group country classification by income level [51].

Search results were uploaded to Rayyan [52]. The project lead (EM) guided two teams of research assistants, two students and four reviewers with lived experience of mental illness in the screening the articles identified in the search. The six reviewers had previously received training on the scoping review process. The reviewers were given access to the identified articles on Rayyan, which they used for screening the articles at both title and abstract levels. The identified articles were divided between the two teams and the three team members of each team then anonymously screened the same articles. Both peer-reviewed and grey literature sources were considered and included if these reported research on supported employment interventions undertaken in LMICs during the last 18 years (2006–2023). Articles on supported employment interventions in LMICs and written in English were included. Conference papers and papers that explicitly exclude mental disorders or psychiatric illness were excluded.

Articles that were provisionally included during screening at title and abstract level, underwent full text review by the same reviewers to confirm inclusion in the review. Disagreements in decisions to include or exclude articles was resolved though consensus.

## Stage 4: Charting the data

The project lead (EM) and a team member (CN) designed a common extraction template capturing basic bibliometric data and main findings. The data was extracted according to key themes related to the study objectives. We then jointly reviewed the full articles extracting relevant information and any discrepancies were resolved through discussion and consensus. We used deductive thematic analysis to synthesize evidence of the supported employment interventions being implemented in LMICs for persons with mental disabilities, with mental disorders such as psychosis, mood disorders and substance use disorders. Stage 5 on collating,

**Table 2. Key terms for database search.**

| Key search terms | Alternative terms | PubMed |
|---|---|---|
| Supported Employment | Individual placement and support, IPS, Place and train model, job coaching, | **Supported employment** [MeSH] |
| Mental health problem | Mental illness *or* Psychiatric disability *or* mental disorders *or* Psychosis *or* substance use disorder or substance abuse | **Mental disorders** [MeSH] (incl. substance related disorders, Substance abuse, Psychosis, substance induced) |

summarising and reporting is presented in the results section. We summarized the data based on the models of supported employment interventions implemented and studied in LMICs and their mental and vocational outcomes for persons with mental disabilities. Furthermore, our reporting was guided by the Preferred Reporting Items for Systematic reviews and Meta-Analyses extension for Scoping Reviews (PRISMA-ScR) Checklist (S1 File).

## Results

### Characteristics of the included studies

The search yielded 9879 studies and after removing duplicates (n = 2532) and screening by title and abstract (n = 7347), 188 studies were assessed for eligibility. The majority (n = 161) were excluded because they were not from LMICs while others were excluded because they were not published in English (2 were in French, 2 in Spanish and 4 in German), were not on mental illness (n = 2) or did not report supported employment interventions (n = 9). At the end, 8 studies were included in the scoping review. The included studies were all peer reviewed articles published between 2015 and 2021, although none were published in 2018 and 2019. Fig 1 summaries the article screening process in a flow chart. Article were from LMICs in Africa [38,53], Asia [26,45,54–56] and Latin America [57]. The articles from Asia where from China [26,54,55] and India [45,56] while both articles from Africa where from South Africa [38,53]. The included studies also included 7 primary studies [26,38,45,53–56] and a critical review [57]. We also reviewed the reference list in critical review and all relevant studies had already been screened for eligibility and excluded. Of the original articles, two were longitudinal descriptive studies [38,53]. The primary studies also included a descriptive retrospective study [56], a feasibility study [45], a case study [55], a qualitative evaluation study [54], and a randomised control trial [26]. Four studies only made mention of supported employment without specifying any particular model of supported employment intervention [38,45,53,56], while 2 specified an ISE model [54,55]. One article made mention of an IPS model [57], while another compared IPS and ISE outcomes [26] Most of the studies suggested involvement of occupational therapists in supported employment interventions [26,38,53–55]. All studies reported populations of persons with severe mental illness including diagnoses of schizophrenia, schizoaffective disorder, bipolar affective disorder, obsessive compulsive disorder, substance use disorders, autism, and intellectual disabilities. Only one study [26] explicitly evaluates and measures symptomatic mental health outcomes, while other two studies [54,55] report on broader mental health outcomes such as improved social relationships, independence, self- efficacy and personal wellbeing among others. Our findings are presented in a typical narrative format with a tabular supplement (Table 3).

### Supported employment interventions available in LMICs

There was very limited research on supported employment interventions are available for persons with mental disabilities in LMICs. The majority of the included studies [38,45,53,56] only referred to a supported employment intervention without mention of a specific type. In those studies that identified the intervention type, Integrated Supported Employment (ISE) [26,54,55] was the most used. Although IPS is embedded in ISE, only one RCT explicitly reported IPS as supported employment intervention in LMICs [26].

The main consideration in IPS is to ensure employment that provides sufficient income to sustain livelihood and that is acceptable to service recipients [54,57]. Cubillos and colleagues [57] suggest that contextual factors may limit the implementation of IPS in LMICs, for example, having to incentivise employers to counteract the perceived risks of employing persons

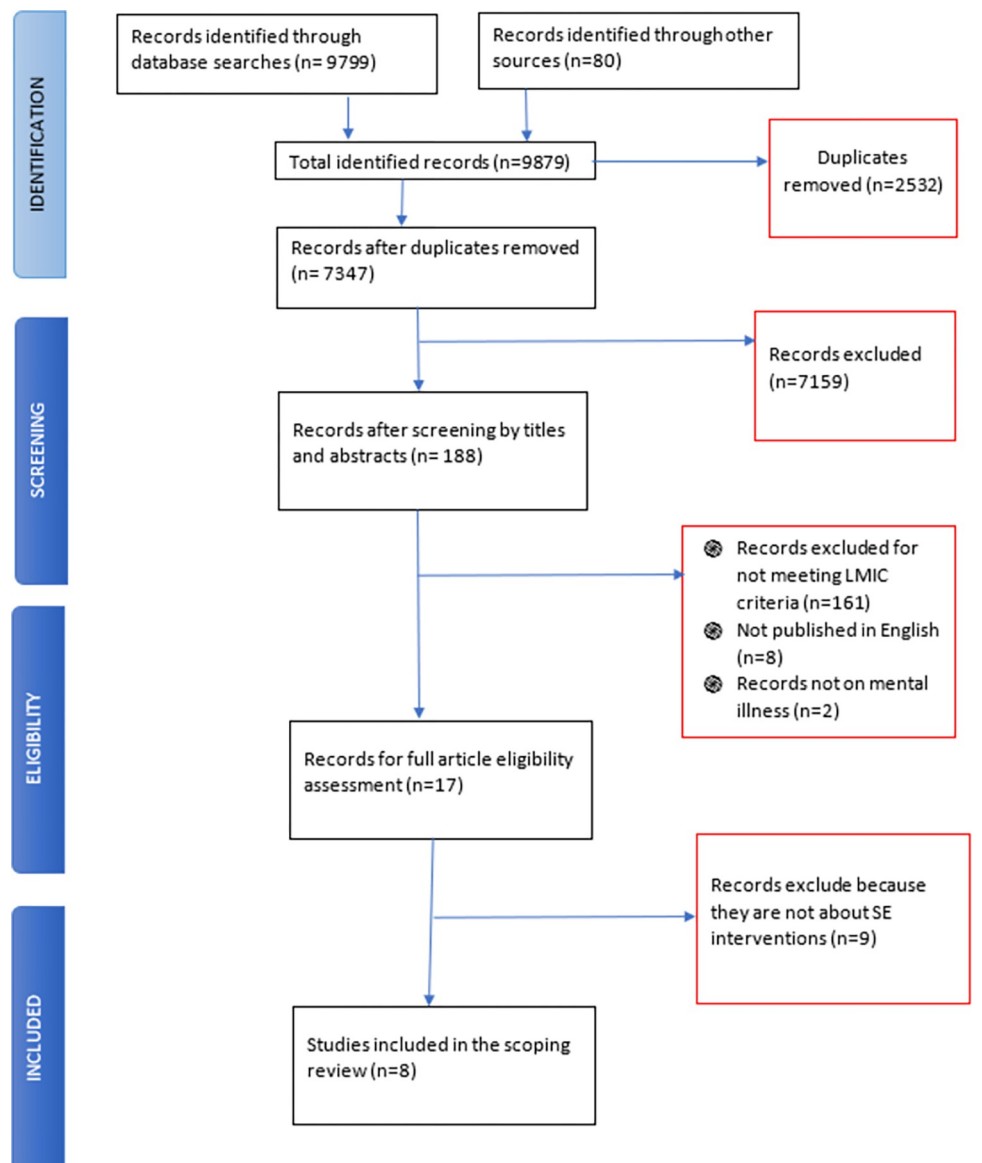

**Fig 1. PRISMA flow chart for screening of articles on supported employment in LMICs.**

with mental disabilities [57]. Other examples are the challenges for reimbursement of service providers through medical aid schemes and the cost implications on public funds [53,57].

## Outcomes of supported employment interventions in LMICs Mental health-related outcomes

Results of supported employment intervention studies in LMICs suggest that supported employment interventions may be an effective way to address various mental health problems in the workplace. Improved mental health outcomes are achieved through providing and sustaining employment for persons with mental disabilities. It is reported that there was a significant reduction in disability and increase in socio-occupational functioning for persons with mental disabilities engaged in supported employment interventions [45]. Socio-occupational

**Table 3. Summary on studies on supported employment interventions in LMICs.**

| | Author(s) (Year) & Country | Model(s) of supported employment intervention(s) studied. | Study design/ Methods | Study population & Sample size (n) | Study aim | Main findings |
|---|---|---|---|---|---|---|
| 1. | Cubillos et al, (2020) [57] Colombia, Costa Rica, Peru | Individual Placement and support (IPS) | Critical Review | Severe mental illness | To reform the mental health care system and advance the employment of people with disabilities in Colombia, Costa Rica, and Peru | Implementation of IPS in Latin America may be hindered by lack of incentives for employers to hire people with severe mental disorders, lack of reimbursement of essential health services by public fund through increased coordination between health and labour ministries for sustainability, and adequate income from employment attained through IPS programmes. |
| 2. | Engelbrecht et al, (2017) [53] South Africa | Supported employment | Longitudinal descriptive study | Intellectual Disability, Schizophrenia, Bipolar I & Schizoaffective n = 29 | To calculate the cost of a Supported Employment service offered to people with mental disabilities | Cost utilisation for psychiatric disability (PD) cohort averaged USD 74.99 on government sessional salary (GSS) rates and USD 141.66on medical aide reimbursement (MAR) rates. Cost of SUPPORTED EMPLOYMENT for intellectual disabilities (ID) cohort averaged USD 137.34 at the GSS rate, and USD 266.69 at the MAR rate per month. Rapid downward progression from average USD 421.85 (GSS rate)/USD 819.14 (MAR) in the first month to USD 11.72 (GSS rate)/USD 22.75 (MAR) in the twelfth month for the PD cohort. Similar trend seen in ID cohort although less pronounced. |
| 3. | Haridas et al, (2021) [56] India | Supported employment | Descriptive Retrospective study | Schizophrenia, Bipolar, OCD and depressive disorders n = 67 | To identify and describe the profile of persons with mental disabilities who had sought the supported employment services | Participants had an increased average duration of employment (37.04 ± 17.29 days) and a 3:1 success-failure ratio for job placement. |
| 4. | Jagannathan et al, (2020) [45] India | Supported employment | Feasibility study | Schizophrenia & Bipolar affective disorder n = 63 | To test the feasibility of a Supported Employment Programme for persons with mental disabilities | 50.8% of participants were successfully placed. The median job tenure was 60 days. Participants were less likely to sustain the job if they faced adverse events (26 days) than if they did not (75 days). Significant changes in disability and socio-occupational functioning in a 6-month period. Disability score significantly reduced (baseline median (IQR)): 5.50 (4.00), 6 months median (IQR): 4.00 (3.00); $z = -2.35$; $p = .02$) and socio-occupational functioning significantly improved (baseline median (IQR): 17.00 (4.00), 6 months median (IQR): 20.00 (8.00); $z = -3.27$; $p = .01$) Reported reduced disability and improved socio-occupational functioning |

(*Continued*)

**Table 3.** (Continued)

| | Author(s) (Year) & Country | Model(s) of supported employment intervention(s) studied. | Study design/ Methods | Study population & Sample size (n) | Study aim | Main findings |
|---|---|---|---|---|---|---|
| 5. | Lu et al, (2015) [55] China | Integrated supported employment (ISE) | Case study | 37-year-old female with paranoid schizophrenia n = 1 | To illustrate the successful implementation of the Integrated Supported Employment (ISE) program in helping people with severe mental illness (SMI) secure continuous employment | Client stayed in employment for 17 months and beyond completion of the intervention Reported improve self-efficacy and improved personal well being |
| 6. | Van Niekerk et al, (2015) [38] South Africa | Supported employment | Longitudinal descriptive study | Intellectual Disability, Schizophrenia, Bipolar I & Schizoaffective n = 29 | To determine the feasibility of an SE service for use in South Africa, specifically as it relates to costs | An average monthly time utilization over 12-months of 9.19 hours with a rapid decrease from 37.22 hours initially to 2.29 hours at the end. Total time utilization higher of Intellectual disabilities (130.83 hours) than for the others (71.38 hours) |
| 7. | Yu et al, (2016) [54] China | Integrated supported employment (ISE) | Qualitative evaluation study | Schizophrenia & Caregivers n = 15 | To explore and compare the views of participants and the caregivers towards the integrated supported employment (ISE) process and outcomes | ISE should help participants to gain employment, be financially independent and have a livelihood. Work-related Social Skills training (WSST) was key to success of ISE. Practical and emotional support also key to ISE success. Gaining and maintaining competitive employment through ISE resulted in improve family communication and family support. Also reported improved mood and improved self confidence |
| 8. | Zhang et al, (2017) [26] China | Integrated supported employment (ISE) Individual Placement and Support (IPS) | Randomised control trial | Schizophrenia n = 162 | The effectiveness of integrated supported employment (ISE) compared with individual placement and support (IPS) and traditional vocational rehabilitation (TVR) for people with schizophrenia | Significantly higher employment rate and longer job tenure were found in the ISE group (63.0%, 29.56 wk) compared with the IPS group (50.0%, 25.47 wk) and TVR group (33.3%, 9.91 wk). The ISE group also attained the most positive psychological outcomes. Reported improved psychosocial functioning and improved subjective Quality of Life |

functioning denotes the ability of an individual to be part of the community in which they reside [58]. Other mental health outcomes related to supported employment interventions include an improved self-reported personal wellbeing, mental health-related recovery and improved self-efficacy through gaining employment leading to financial independence [45,54,55]. Beneficiaries of supported employment programmes and interventions further report improved social skills, improved stress coping mechanisms, enhanced independent living skills, functional restructuring of daily activities and improved drive to seek and attain work [45]. Additionally, supported employment interventions also improved communication in families of service users, with improvements of caregiver attitudes towards individuals with severe mental disorders [45,54]. Furthermore, supported employment intervention showed higher functional assessment measure scores, related to mental health outcomes, on the Global Assessment of Functioning (GAF) and the Personal Wellbeing Index (PWI) than TVR [26]. For example, those who participated in the ISE programme scored a mean of 65.83 and 53.37

on the GAF and PWI respectively compared with means of 62.50 and 25.46 on the GAF and PWI respectively in the group receiving TVR [26]. This suggests that the group receiving ISE had higher level of psychosocial functioning and subjective quality of life than the TVR group.

**Vocational outcomes.** Supported employment interventions have been reported to be more effective in both vocational and functional outcomes than traditional vocational rehabilitation (TVR) approaches, like the train-and-place model, in LMICs [26]. A randomised control trial by Zhang and colleagues [26] reported that supported employment interventions resulted in better vocational outcomes, for example employment rates and job tenure. Up to 63% of participants in supported employment intervention groups gained employment and stayed in employment for up to an average of 29.56 weeks, compared to 33% in the TVR group with an average employment duration of 9.91 weeks. Furthermore, among the supported employment intervention groups there were also significant differences, with ISE having better vocational outcomes than IPS (50% employment rate, 25.47 weeks job tenure, p-value = 0.02) and significantly higher functional scores for most measures used [26].

Supported employment interventions have been reported to show increased job attainment, tenure and sustainability for persons with mental disabilities in LMICs [26,38,45,53,55]; these have been reported as key social determinants of mental health with a positive influence on mental health outcomes [59]. However, the studies under review did not explore the relationship between employment, as a social determinate of mental health, and mental health outcomes. Supported employment interventions resulted in good job placement success rates ranging from 50.8–63% [26,45]. Haridas and colleagues [56] reported that 3-in-4 individuals from a tertiary rehabilitation institution who sought for employment were successful. With supported employment interventions, persons with mental disabilities could sustain employment from between 37 days and 60 days [45,56]. The duration of this job tenure was greatly affected by prior training, facing hardship in the work place and support received (duration and type) [45]. In a case study by Lu and colleagues [55] in Mainland China, the individual was able to attain and sustain employment beyond 17 months with appropriate support and requisite training.

Findings of a modest prospective study that captured the time utilization and cost of supported employment interventions reported a rapid decrease after the first month [38,53]. Furthermore, the findings suggest supported employment interventions might be a cost-effective service, with an average monthly time utilisation of 9.19 hours over a 12-month period, meaning service users actively utilized 9.19 hours of support during an average of 1920 work hours [38]. In addition, the promising cost effectiveness of supported employment interventions based on time utilisation of individual service users, can further be enhanced in cases where service components being access through group formats, which is the case in 70% of the services offered to persons with Intellectual Disability [38,53].

## Discussion

There is limited research evidence of supported employment interventions from LMICs. All included studies were from three two upper-middle income countries, China and South Africa, and a low-middle income country, India. In addition, a critical review focussed on three upper-middle income countries, Costa Rica, Colombia and Peru. The limitation of the findings might be due to the shifting economic fortunes in these countries as their economies are fast growing with increased opportunities for employment [55]. Without adequate financing and resourcing, supported employment interventions are economically unsustainable [60], therefore may be challenging to implement in lower income countries. Furthermore, the socio-political environment in these countries have a positive impact on development of

supported employment interventions, as they have established pro-disability employment laws and policies [47].

Studies from India and South Africa only make mention of a supported employment interventions while studies from China explicitly name IPS and ISE. This suggests that increasing economic stability promotes implementation of more robust evidence-based models beyond the generic supported employment model. Rusch and Hughes [61] explain that supported employment is a group of interrelated work intervention models that emphasise integrated paid work with ongoing support. The most widely researched model of supported employment intervention model is the IPS [25,41,62–64]. ISE was developed in Hong Kong in response to the contextual limitations on implementing IPS to fidelity and to improve job tenure and sustaining employment outcomes [28,65] Supported employment interventions have increasingly integrated other interventions and services such as cognitive interventions, social skills interventions, clubhouse approaches, sheltered employment and occupational therapy to improve outcomes for persons with mental disabilities and in response to contextual realities [66].

The ISE approach was shown to be more effective than IPS in achieving both employment and mental health outcomes in LMICs. This might have been because the ISE approach was an augmented IPS model with additional elements to promote social and interpersonal functioning in the workplace [65,67]. Zhang and colleagues [26] highlight that the ISE developed from evidence-based occupational therapy interventions. The influence of occupational therapists and occupational therapy in supported employment interventions is a recurring theme in literature [68–75]. Occupational therapists apply their competencies in mental health and work rehabilitation to promote social inclusion, recovery and open labour market participation by persons with mental disabilities [76,77]. Occupational therapists address the work and employment needs of persons with mental disabilities through the use their knowledge of work and skills in occupational analysis, contributing to the design and implementation of supported employment interventions [27,71]. Furthermore, occupational therapists utilise a range occupation-based assessment in evaluating the success and impacts of supported employment interventions [71,73]. Abidin and colleagues [78] suggest that occupational therapists should implement ISE as it permits a more holistic approach, in line with occupational therapy philosophy and practice, resulting in better mental and vocational outcomes.

Although evidence is currently limited, available literature suggests supported employment interventions have been implemented with success in LMICs [26,45,54,55]. It is noteworthy that despite mounting evidence of the successes and benefits of supported employment interventions, such as the IPS approach [32,40,79,80], implementation fidelity in LMICs is often low as implementors respond to contextual realities that may limit success of the approach [45,57]. Various adaptations to supported employment interventions have been applied including pre-employment social skills training, integration into existent mental health services and extended support for both service users and employers to facilitate its implementation success in LMICs [27,45,55]. ISE services were reported across various LMICs and provided a good alternative to the IPS generally implemented in high income countries. ISE was reported to improve social functioning and interpersonal relationships in the workplace, which in turn led to improved vocational outcomes, consistent with its intended purpose [28,78].

The current evidence suggests that supported employment interventions are effective in achieving employment outcomes among persons with mental disabilities in LMICs. Supported employment interventions resulted in significantly higher placement rates in competitive employment and longer duration in employment [26,27]. Furthermore, supported employment interventions showed enhanced work attainment for persons with mental disabilities

and increased job tenure to the benefit of both persons with mental disabilities and their families [54]. Above half of participants in the studies reviewed were able to attain employment and could retain employment with appropriate support [26,27,55]. Supported employment interventions has been shown to improve on both job placements and job tenure for persons with mental disabilities as compared to traditional train-and-place approaches [32,68,81–83].

Participation in work has been seen to improve mental health outcomes. Although there is limited evidence of the effectiveness of supported employment interventions on mental health outcomes in the studies under review, literature suggests that employment positively improves mental health outcomes [29,76,84]. Moreover, the reviewed studies had short-term follow-up periods, while literature suggests that in the long-term duration of employment may yield and sustain positive mental health outcomes over time [59,85,86]. The limited research evidence on mental health benefits of employment in LMIC confirm the need for further research.

[76,77]. The results highlighted that supported employment interventions are a cost-effective approach to employment for persons with mental disabilities despite the high costs in the inception stages. This resonated with findings from other settings which highlighted that supported employment interventions become increasingly cost effective over time [43].

Major gaps in evidence remain in low-income countries. All of the identified studies in this scoping review were from upper and lower middle-income countries. Therefore, there is a gap in the evidence of implementation of supported employment interventions in low-income countries, as well as their effectiveness in these contexts. Furthermore, only a single research group reported on mental health outcomes, with only one study attempting to measure mental health outcomes among participants of supported employment interventions in LMICs. This limited the extent of literature from which we could draw insights and inferences on the true effectiveness of supported employment interventions and makes it difficult to generalise findings from a single study.

## Conclusions

The scoping review was undertaken to synthesize research on supported employment interventions undertaken in LMICs and consider the importance of supporting decision making for implementation this service in the workplace. There is a dearth of research evidence of supported employment interventions from LMICs. Therefore, there is need for further research on their implementation and outcomes in LMICs. The available evidence suggests supported employment interventions are beneficial to realisation of vocational and employment needs of persons with mental disabilities. Supported employment interventions are cost-effective and if embedded in existing mental health services, led-by and involving occupational therapists, may lead to best outcomes.

### Strengths and limitations

➢ The scoping review methodology is a robust methodology to not only synthesize the available literature, but also to identify gaps in the available evidence.

➢ More than half of the identified studies were research conducted by two research groups. This resulted in limitations in identification of the types of supported employment interventions that are available in LMICs.

➢ The exclusion of sources not written in English could have resulted in relevant studies on supported employment interventions from French-speaking or Spanish-speaking LMICs being missed.

➢ The included studies were not subjected to a quality appraisal.

## Supporting information

**S1 File. PRISMA-ScR checklist.** Preferred Reporting Items for Systematic reviews and Meta-Analyses extension for Scoping Reviews (PRISMA-ScR) Checklist.
(DOCX)

## Acknowledgments

Special thanks go to lived experience experts that reviewed and gave their insights to our scoping review and analysis.

## Author Contributions

**Conceptualization:** Edwin Mavindidze, Clement Nhunzvi, Lana Van Niekerk.

**Formal analysis:** Edwin Mavindidze, Clement Nhunzvi, Lana Van Niekerk.

**Funding acquisition:** Edwin Mavindidze, Clement Nhunzvi, Lana Van Niekerk.

**Investigation:** Edwin Mavindidze.

**Methodology:** Edwin Mavindidze, Clement Nhunzvi, Lana Van Niekerk.

**Project administration:** Edwin Mavindidze.

**Visualization:** Edwin Mavindidze.

**Writing – original draft:** Edwin Mavindidze, Clement Nhunzvi.

**Writing – review & editing:** Edwin Mavindidze, Clement Nhunzvi, Lana Van Niekerk.

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
