## [Decision Letter · Decision Letter 0]

22 Jun 2023

PONE-D-23-15544Supported employment interventions for workplace mental health of chronic mental service users in low-to-middle income countries: a scoping reviewPLOS ONE

Dear Dr. Mavindidze,

Thank you for submitting your manuscript to PLOS ONE. After careful consideration, we feel that it has merit but does not fully meet PLOS ONE’s publication criteria as it currently stands. Therefore, we invite you to submit a revised version of the manuscript that addresses the points raised during the review process.

We look forward to receiving your revised manuscript.

Kind regards,

Amin Yazdani

Academic Editor

PLOS ONE

Journal Requirements:

Reviewers' comments:

Reviewer's Responses to Questions

**Comments to the Author**

1. Is the manuscript technically sound, and do the data support the conclusions?

Reviewer #1: Yes

Reviewer #2: Yes

2. Has the statistical analysis been performed appropriately and rigorously? 

Reviewer #1: Yes

Reviewer #2: N/A

3. Have the authors made all data underlying the findings in their manuscript fully available?

Reviewer #1: Yes

Reviewer #2: Yes

4. Is the manuscript presented in an intelligible fashion and written in standard English?

Reviewer #1: Yes

Reviewer #2: Yes

5. Review Comments to the Author

Reviewer #1: Thank you for the opportunity to review this paper. The authors reviewed the evidence of supported employment in low-to-middle income countries, its effectiveness on mental health outcomes through the promotion of the participation fmental health service users in the open labour market.

Overall, this is a well written study that is needed. The state of support employment in LMIC is relatively unexplored in the workplace mental health literature. I invite the authors to add additional contexts in the background section about the dearth of literature as they further develop the need for this study by identifying the research gaps. Moreover, there should be a more fulsome discussion of occupational therapy in addressing support employment needs in LMIC countries. This is relatively unexplored in the study, although the authors mention a few studies in lines 202 and 206.

Table 2 also identifies studies from LMICs, however, I question how the study from Tan et al., 2016 (Singapore) would be considered a LMIC. Please expand on this in the study.

The discussion section should also be further expanded and aligned with the extant literature. How does the existing scoping review findings enhance our understanding of supported employment. The limitation section also needs to be further elaborated in addition to the conclusion.

I look forward to reviewing the next draft of the paper

Reviewer #2: Thank you for your efforts in conducting this important research to synthesize the existing literature on supported employment interventions in low-middle income countries, with a focus on how they impact persons with mental disabilities. It is a large undertaking to screen over 7000 title abstracts, full-text review over 150 articles. For the most part, the scoping review is well-structured; however, there are also sections that are unnecessarily wordy and inconsistent which may lead to confusion. This manuscript requires significant revisions prior to resubmission for publication.

The comments below are intended to strengthen and clarify the manuscript.

Overall comments:

The study of objectives must be clarified and remain consistent throughout the manuscript.

The authors repeat the study objectives in several sections of the manuscript (i.e., line 4, 25, 114, 120), but it is not done in a consistent nor concise manner. The study objectives must be clarified as they shape the entire paper. First, it is not recommended that the objective is to “moot” or “advocate” for a predetermined outcome; this is NOT one of the purposes of scoping reviews (Arskey & O’Malley, 2005). Perhaps softer language like, “support decision-making” may be used. Second, it is recommended that the term “interventions” be constantly added to “supported employment”. “Supported employment” and “supported employment interventions” may not mean the same thing. “Supported employment” may be interpreted as a specific intervention, whereas “supported employment interventions” may be interpreted as a more general type of intervention. Third, mapping the evidence may require figures such as a link diagram. Based on what was presented in the manuscript, perhaps the term “synthesize” would be more appropriate. Lastly, it is unclear whether the scope of the review pertains to the impacts of supported employment interventions on persons with mental disorders OR the mental health outcomes associated with supported employment interventions. Since the review excluded articles that “explicitly exclude mental disorders or psychiatric illness,” and included outcome measures beyond those that are directly related to mental health (i.e., cost, service usage), the authors should clarify that the scope of the review ‘synthesizes the existing literature on supported employment interventions in low-middle income countries, and documents their impact for persons with mental disabilities.’ In a similar vein, it is recommended that the authors remove the term “(chronic) mental health service users.” This is because the authors have not verified that all participants in the retained articles are using mental health services, rather, they were defined by their mental illness (as demonstrated in table 2, study population). Perhaps using the term “persons with mental disabilities” would be more accurate. Please make the appropriate changes throughout the manuscript to address the concerns and suggestions identified above to concisely frame the scope and objectives of the review and avoid confusions for the reader.

Please make mention of all figures and supplementary materials in-text so they are not independent from the manuscript (i.e., figure 1, supplementary materials)

Please review the article for typos, and grammatical errors (particularly use of commas). Also, please be consistent in the use of acronyms.

Please review the article as a whole to ensure that it is consistent with the study objectives.

Please ensure abstract reflect the changes made to the main text.

Specific Comments by Section

Introduction:

Second paragraph: Including a sentence to describe the Waddel & Burton (2006) report would strengthen the argument that work is overall beneficial: Waddell, G., & Burton, A. K. (2006). Is work good for your health and well-being?

Line 75: include the term ‘adverse’ mental health outcomes like depression and anxiety.

Link 89-93: This paragraph is out of place and may not be necessary. It shifts the focus to workplace mental health promotion and does not support the importance of supported employment interventions. This leaves the reader questioning “How can employers improve workplace mental health promotion,” rather than focussing on the benefits/importance/potential of supported employment interventions. The Waddell & Burton (2006) report suggests that work is overall good for health despite the stresses that it comes with. It may be worth acknowledging the stressors of work in paragraph 2, but emphasize the work is overall good for health.

Line 94-104: Please describe the various models of supported employment interventions, including the Clubhouse-model (line 108) and the ones defined in Table 3.

Line 114-117”: This sentence is unnecessarily complicated. Suggestion: “This scoping review aims to synthesize the existing literature pertaining to supported employment interventions in LMICS and document their impacts on persons with mental disabilities. Findings may be used to support decision-makers in the development, implementation, and execution of supported employment interventions.”

Methods:

Line 119: If this review adopted the Arksey and O’Malley framework, please use it. It is unclear why the authors referenced the Arksey and O'Malley framework and then summarize the methods using the SALSA framework, and attaching the (PRISMA-ScR) Checklist (2018) in the supplemental materials. The Arksey and O’Malley framework consists of five stages: (1) identify the research questions; (2) search for relevant studies; (3) select relevant studies; (4) chart the data; and (5) collate, summarize, and report the results. Please be consistent with one framework to avoid confusion.

Line 122: incomplete phrase � supported employment for ______ and gauge….

Please specify when the search was conducted.

Line 123: The research questions need to reflect the aim of the scoping review. The research question “What supported employment interventions are available for adults with chronic mental health problems in LMIC?” is too broad and the review does not actually aim to identify supported employment interventions are available for adults with chronic mental health problems in LMIC; rather it synthesizes the literature. Therefore, I believe the research questions are “What is the existing literature/research on supported employment interventions in LMIC?” and “What is the impact of supported employment interventions for persons with mental disabilities in LMIC?”

Line 162 states: “extracted according to key themes related to the study objectives”. Please expand on how inductive thematic analysis (line 164) was applied. Extracting data according to key themes developed a priori is NOT inductive thematic analysis.

Line 164: Suggest to write “We summarized the data based on the models of supported employment interventions implemented and studied in LMICs and their mental and vocational outcomes for persons with mental disabilities.

Line 168-179: The last two sentences should be in the results.

Results:

Table 2: A few recommendations to reduce the length and size of the table:

1. Combine authors, year, and country in the same column. “Country, Author(s), (Year)”.

2. Combine mental health outcomes and main findings into the same column.

Please include sample size in the study population in Table 2.

Please change intervention to Model(s) of supported employment intervention(s) studied in Table 2.

Line 186: Please clarify that the authors reviewed the reference list of the Critical Review that was retained in the review. It is typically preferred to include the primary research rather than reviews.

Line 197: Since this section specifies SE interventions models that have been studied, it should include the frequency of each model that was studied. In other words, how many studies researched the WSST model, the IPS model, and the ISE model in LMICs. How many studies did not specify the model?

Line 198: The research methods used cannot determine that there are very limited supported employment interventions available for mental health service users in LMICs. Limited research on this topic does not imply that the programs do not exist. It is recommended that “There was very limited research on….” be added at the beginning of the Results section. Otherwise, please justify how the systematic search and inclusion criteria allowed for this statement.

Line 209: Table 3 belongs in the introduction. The definitions of WSST IPS, and ISE did not come from the included articles. This is not a theme or a finding from the retained articles.

Line 218: Suggestion to change subtitle to Outcomes of Supported Employment Interventions and to have to smaller headers: 1) Mental health-related outcomes; 2) Vocational outcomes.

Line 245-252: Move to mental health outcomes (i.e., GAF and PWI) to previous paragraph.

Line 277: This is incorrect: month time utilisation is 9.19 (i.e., 9.19 hours/month). Therefore, usage is 110 hours per year (9.19 hours/month x 12 months).

Please clarify if the 6 reviewers screened all 7347 articles, or if the articles were divided amongst the two teams, did at least reviewers read over each title and abstract?

Please provide a reference for how LMIC was defined.

Line 154: Please include ‘written in English’ as an inclusion criteria.

Discussion

The thoughts and points brought forth in the discussion were scattered and difficult to read. This section requires significant revisions. It is suggested that the authors refer back to the research questions to guide the discussion. Below are some additional suggestions:

First paragraph: It is unclear how this paragraph is related to the study findings? Please help the reader connect the points. Otherwise, this paragraph should be reframed and placed in the Introduction to provide rationale for a scoping review on supported employment interventions in LMICs.

Similarly, please summarize the main results from this review and connect the discussion back to the two research questions.

It is recommended that the authors discuss SE in general before discussing the nuances and benefits of specify SE approaches.

Line 300 states that evidence is currently limited, then Line 301 states that there is mounting evidence of the successes and benefits of an IPS approach in LMICs. This is confusing.

Line 296: Please describe what is meant by implementation support rather than stating “implementation support is key to successful implementation”. Is it funding, expertise, personnel, infrastructure, etc.? Please be more specific.

Line 312: The phrase, “has been established,” implies conclusive findings. It is difficult to make that conclusion when only 9 relevant articles were identified. Further, a quality appraisal of the research was not conducted. Please rephrase.

Line 302: References 24, 32, 55, 56 were not retained for the scoping review. Please clarify why references 24, 32, 55, 56 were used to support the fact that SE interventions have been implemented in LMICs.

Line 324: Reference 62 and 63 were not one of the nine retained studies. Please rephrase or remove the sentence.

Line 356: Please clarify how the two research group that conducted most of the studies retained in this review limited the identification of other types of supported employment available in LMICs? It seems that it is simply a lack of research in this area.

Line 359: This is not a limitation of your review. However, it is a research gap. This point can be included with the paragraph at line 341. Also, it would be worthwhile to discuss/explore why there are so few studies conducted on this topic and the barriers to conducting research on SE in LMICs. It would also be fruitful to discuss the barriers and facilitators to implementing SE in LMICs.

6. PLOS authors have the option to publish the peer review history of their article (what does this mean?). If published, this will include your full peer review and any attached files.

Reviewer #1: No

Reviewer #2: No

---

## [Author Response · Author response to Decision Letter 0]

9 Aug 2023

All the reviewer comments have been addressed and responded in tabular format in a document submitted with this review.

---

## [Decision Letter · Decision Letter 1]

7 Sep 2023

Supported employment interventions for workplace mental health of persons with mental disabilities in low-to-middle income countries: a scoping review

PONE-D-23-15544R1

Dear Dr. Mavindidze,

We’re pleased to inform you that your manuscript has been judged scientifically suitable for publication and will be formally accepted for publication once it meets all outstanding technical requirements.

Kind regards,

Amin Yazdani, PhD

Academic Editor

PLOS ONE

Additional Editor Comments (optional):

Reviewers' comments:

Reviewer's Responses to Questions

**Comments to the Author**

1. If the authors have adequately addressed your comments raised in a previous round of review and you feel that this manuscript is now acceptable for publication, you may indicate that here to bypass the “Comments to the Author” section, enter your conflict of interest statement in the “Confidential to Editor” section, and submit your "Accept" recommendation.

Reviewer #1: All comments have been addressed

Reviewer #2: All comments have been addressed

2. Is the manuscript technically sound, and do the data support the conclusions?

Reviewer #1: Yes

Reviewer #2: Yes

3. Has the statistical analysis been performed appropriately and rigorously? 

Reviewer #1: N/A

Reviewer #2: N/A

4. Have the authors made all data underlying the findings in their manuscript fully available?

Reviewer #1: Yes

Reviewer #2: Yes

5. Is the manuscript presented in an intelligible fashion and written in standard English?

Reviewer #1: Yes

Reviewer #2: Yes

6. Review Comments to the Author

Reviewer #1: (No Response)

Reviewer #2: This is a much stronger manuscript. Thank you for making the suggested revisions. No further comments.

7. PLOS authors have the option to publish the peer review history of their article (what does this mean?). If published, this will include your full peer review and any attached files.

Reviewer #1: No

Reviewer #2: No

---

## [Editor Report · Acceptance letter]

13 Sep 2023

PONE-D-23-15544R1 

 Supported employment interventions for workplace mental health of persons with mental disabilities in low-to-middle income countries: a scoping review 

Dear Dr. Mavindidze:

I'm pleased to inform you that your manuscript has been deemed suitable for publication in PLOS ONE. Congratulations! Your manuscript is now with our production department. 

Kind regards, 

on behalf of

Dr. Amin Yazdani 

Academic Editor

PLOS ONE